# Temporally integrated single cell RNA sequencing analysis of PBMC from experimental and natural primary human DENV-1 infections

Adam T. Waickman[1,2,3]*, Heather Friberg[1], Gregory D. Gromowski[1], Wiriya Rutvisuttinunt[1], Tao Li[1], Hayden Siegfried[1], Kaitlin Victor[1], Michael K. McCracken[1], Stefan Fernandez[4], Anon Srikiatkhachorn[5,6], Damon Ellison[1], Richard G. Jarman[1], Stephen J. Thomas[2], Alan L. Rothman[5], Timothy Endy[3], Jeffrey R. Currier[1]

1 Viral Diseases Branch, Walter Reed Army Institute of Research, Silver Spring, Maryland, United States of America, 2 Institute for Global Health and Translational Sciences, State University of New York Upstate Medical University, Syracuse, New York, United States of America, 3 Department of Microbiology and Immunology, State University of New York Upstate Medical University, Syracuse, New York, United States of America, 4 Department of Virology, Armed Forces Research Institute of Medical Sciences, Bangkok, Thailand, 5 Department of Cell and Molecular Biology, Institute for Immunology and Informatics, University of Rhode Island, Providence, Rhode Island, United States of America, 6 Faculty of Medicine, King Mongkut's Institute of Technology Ladkrabang, Bangkok, Thailand

* waickmaa@upstate.edu

**Data Availability Statement:** All data supporting the findings of this study are available within this article and its Supplementary Information files. Single-cell RNAseq gene expression data have

## Abstract

Dengue human infection studies present an opportunity to address many longstanding questions in the field of flavivirus biology. However, limited data are available on how the immunological and transcriptional response elicited by an attenuated challenge virus compares to that associated with a wild-type DENV infection. To determine the kinetic transcriptional signature associated with experimental primary DENV-1 infection and to assess how closely this profile correlates with the transcriptional signature accompanying natural primary DENV-1 infection, we utilized scRNAseq to analyze PBMC from individuals enrolled in a DENV-1 human challenge study and from individuals experiencing a natural primary DENV-1 infection. While both experimental and natural primary DENV-1 infection resulted in overlapping patterns of inflammatory gene upregulation, natural primary DENV-1 infection was accompanied with a more pronounced suppression in gene products associated with protein translation and mitochondrial function, principally in monocytes. This suggests that the immune response elicited by experimental and natural primary DENV infection are similar, but that natural primary DENV-1 infection has a more pronounced impact on basic cellular processes to induce a multi-layered anti-viral state.

been deposited in the Gene Expression Omnibus database (GSE154386). link: https://www.ncbi.nlm.nih.gov/geo/query/acc.cgi?acc=GSE154386.

**Funding:** This work was supported by the Military Infectious Disease Research Program (AW), the Congressionally Directed Medical Research Program (JRC), and the National Institutes of Allergy and Infectious Disease (NIAID, P01AI034533, ALR). The funders had no role in study design, data collection and analysis, decision to publish, or preparation of the manuscript.

**Competing interests:** I have read the journal's policy and the authors of this manuscript have the following competing interests:A.T.W reports grants from Military Infectious Disease Research Program, during the conduct of the study. A.L.R. reports grants from National Institute of Allergy and Infectious Diseases, during the conduct of the study. S.J.T reports other support from US DoD, other support from GSK, during the conduct of the study; personal fees and other support from GSK Vaccines, personal fees and other support from Takeda, personal fees and other support from Merck, personal fees and other support from PrimeVax, personal fees and other support from Themisbio, personal fees and other support from Chugai Pharma, personal fees and other support from Cormac Life Sciences, personal fees and other support from HHS NVPO / Tunnel Govt Services, personal fees and other support from Janssen, other support from GreenMark Partners. In addition, S.J.T has a patent US10086061B2 (combined flavivirus vaccines) issued. J.R.C reports grants from the Congressionally Directed Medical Research Program during the conduct of the study. All other authors have nothing to disclose.

## Author summary

Dengue Human Challenge Models allow for the analysis of host/virus interactions under highly controlled experimental conditions. However, it is unclear how close the immune response generated by an attenuated challenge virus compares to that generated by a naturally acquired DENV infection. In this study, we utilized single cell RNA sequencing to assess the immune response generated by both experimental and natural primary DENV-1 infections. This analysis suggests that the immune response elicited by experiential and natural primary DENV-1 infections are similar, but that natural DENV-1 infection has a more pronounced impact on basic cellular processes to induce a multi-layered anti-viral state.

## Introduction

Dengue is one of the most widespread vector-borne viral diseases in the world. The causative agent–dengue virus (DENV)–is a positive-stranded RNA virus maintained in an anthroponotic cycle between the *Aedes aegypti* mosquito and humans [1]. Consisting of four co-circulating but immunologically and genetically discrete serotypes (DENV-1, -2, -3, and -4), DENV is thought to infect up to 300 million individuals yearly [2,3]. Although the majority of DENV infections are subclinical, as many as 100 million infections every year result in symptomatic dengue fever. In its most severe manifestation, dengue fever can progress to dengue hemorrhagic fever/dengue shock syndrome (DHF/DSS) [4–7]. While the pathogenesis of severe dengue is complex and may involve some degree of genetic predisposition, severe symptoms are more likely to occur in individuals previously infected with a heterologous viral serotype compared to individuals without any preexisting DENV immunity [8,9]. Despite decades of study, the precise mechanisms underpinning this unique epidemiological feature of DENV infection remain unresolved and continues to impede the development of an effective DENV vaccine.

In addition to the unclear role of preexisting immunity and other host/environmental factors on the immunopathogenesis of DENV infection, the fundamental immunological and molecular signatures associated with early DENV infection remain mostly unresolved. This is partially attributable to the endemic nature of DENV–resulting in frequent and unpredictable exposures in susceptible individuals -as well as the wide variation in the kinetics of asymptomatic incubation/infection prior to any symptomatic manifestation of infection [10]. Resolving the early unique immunological and molecular signatures specific to DENV infection would aid in the development of rapid diagnostic tools, as well as provide insight into the basic pathophysiology of infection and disease.

DENV human infection models (DHIMs) offer a unique opportunity to formally address many of the outstanding questions in the field of DENV biology. By exposing pre-screened individuals to DENV in a controlled setting, it is possible to closely monitor the immunological response elicited by infection prior to the onset of any traditional clinical symptoms. In addition to providing a platform to answer many fundamental questions of flavivirus biology, DHIMs are a powerful tool to systematically develop and down-select candidate vaccine platforms or therapeutic agents in a controlled setting before expanding to large-scale efficacy trials. Indeed, Kirkpatrick and colleagues have demonstrated the utility of this approach by utilizing an attenuated DENV-2 challenge virus to test the efficacy of the TV003 DENV vaccine product prior to the initiation of larger efficacy trials [11].

While the earliest DHIM studies performed in the 1930s utilized unattenuated wild-type DENV strains, all current DHIM studies utilize highly characterized and attenuated viral

strains. Viral attenuation is critical to ensure an acceptable safety profile, but it leaves open the possibility that the causative mutations may significantly alter the immunological response elicited by infection, thereby reducing the utility of the model. Limited data are available on how closely these experimental primary DENV infections mimic the immunological subtleties and complexities of natural primary DENV infection. The majority of DHIMs appear to recapitulate the basic clinical features associated with mild natural primary DENV infection, including rash, myalgia, headache, and fever, as well as the crude virologic/serologic features of infection such as peripheral viremia/RNAemia and DENV-specific seroconversion [12,13]. However, no study to date has attempted to directly compare/contrast the molecular signatures associated with either natural or experimental primary DENV infection with any degree of cellular and/or temporal resolution. Understanding how closely an experimental primary DENV infection mimics the molecular subtleties of a wild-type DENV infection would inform decisions on how best to utilize the tool for product down-selection, as well as provide insight into the basic molecular response elicited by primary DENV infection with an unparalleled degree of temporal resolution.

To close this significant knowledge gap, we utilized high-throughput single-cell RNA sequencing (scRNAseq) technology to assess the longitudinal transcriptional profile associated with both experimental and natural primary DENV-1 infection with single-cell resolution. This study utilized samples collected as part WRAIR/SUNY phase one open label DENV-1 human challenge study performed in Syracuse, NY [12], as well as samples collected from children enrolled in a hospital-based acute dengue study in Bangkok, Thailand [14]. Unfractionated PBMC from 8 time points (days 0, 2, 4, 6, 8, 10, 14/15 and 28) from 3 individuals enrolled in the experimental primary DENV-1 infection study were analyzed, as well as 3 time points (acute 1, acute 2, day 180) from two individuals experiencing a natural primary DENV-1 infection. This temporally integrated analysis resulted in the capture of 171,208 cells, which upon examination collapsed into 22 statistically distinct populations corresponding to all major anticipated leukocyte subsets. While all annotated cell populations demonstrated significant and consistent perturbations in their transcriptional profile in response to either natural or experimental primary DENV-1 infection, conventional monocytes respond most robustly to infection across all subjects and study groups from an unbiased transcriptional perspective. Using these data, conserved Differentially Expressed Genes (cDEGs) induced or suppressed by natural or experimental primary DENV-1 were identified, and the overlap between the two arms of the study assessed. The infection-induced cDEGs associated with experimental primary DENV-1 infection were found to reflect a subset within the larger gene set associated with natural primary DENV-1 infection, primarily corresponding to gene products associated with a cellular response to systemic inflammation and interferon (IFN) production. In contrast, the number of infection-suppressed cDEGs was higher in cells obtained following natural primary DENV-1 infection than in cells obtained following experimental primary DENV-1 infection. Infection-suppressed cDEGs primarily corresponded to gene products associated with protein translation/elongation and mitochondrial function, two cellular processes known to be suppressed by prolonged IFN signaling. These results are consistent with the concept that the immune response elicited by an experimental primary DENV-1 infection represents a tempered version of that generated in response to a natural primary DENV-1 infection, and that the more pronounced inflammation associated with natural primary DENV-1 infection has a correspondingly more pronounced impact on basic cellular processes to induce a multi-layered/systemic anti-viral state. These data provide insight into the molecular level response to primary DENV-1 infection, and how viral pathogenesis correlates with immune activation and cellular pathophysiology.

## Results

### Subject selection and characterization

The primary objective of this study was to determine the kinetic transcriptional signature associated with experimental primary DENV-1 infection and to determine how closely this profile correlates with the transcriptional signature accompanying natural primary DENV-1 infection. To this end, three subjects from the SUNY/WRAIR DENV-1 DHIM study were selected for analysis based on their representative viremia and seroconversion kinetics relative to the SUNY/WRAIR DHIM-1 study as a whole (**Fig 1A**) [12]. This human challenge study utilized the 45AZ5 DENV-1 virus strain, which is an under-attenuated DENV-1 vaccine candidate generated by serial passage of the parental Nauru/West Pac/1974 DENV-1 isolate through diploid fetal rhesus lung cell line (FRhL) in the presence of 5-azacytidine [15,16]. This *in vitro* attenuation process resulted in the introduction of 25 nucleotide and 10 amino acid substitution in the attenuated 45AZ5 strain relative to the parental Nauru/West Pac/1974 DENV-1 isolate, with substitutions observed in the E, NS3, NS4a, and NS5 genomic regions of the virus [15].

All subjects included in this study received $3.25 \times 10^3$ PFU of the 45AZ5 DENV-1 challenge virus strain following extensive pre-screening to ensure an absence of preexisting DENV immunity. All three subjects exhibited significant DENV-1 RNAema between 5 and 15 days post-challenge and demonstrated classic staggered IgM/IgG seroconversion between study days 13 and 16 (**Figs 1A and S1**). PBMC from a total of 8 time points per subject were selected for this study, corresponding to study days 0, 2, 4, 6, 8, 10, 14/15, and 28. In addition, PBMC from two children experiencing serologically-confirmed natural primary DENV-1 infections were selected for analysis (**Figs 1B and S1 and S1 Table**). A total of 3 time points per subject were analyzed: two sequential samples collected during the acute (febrile) phase of infection, and a reference sample collected 6 months post-defervescence. For both the experimental and natural primary DENV-1 infection subjects, each sample was analyzed in parallel by scRNAseq and by flow cytometry (**S1, S2 and S3 Figs**).

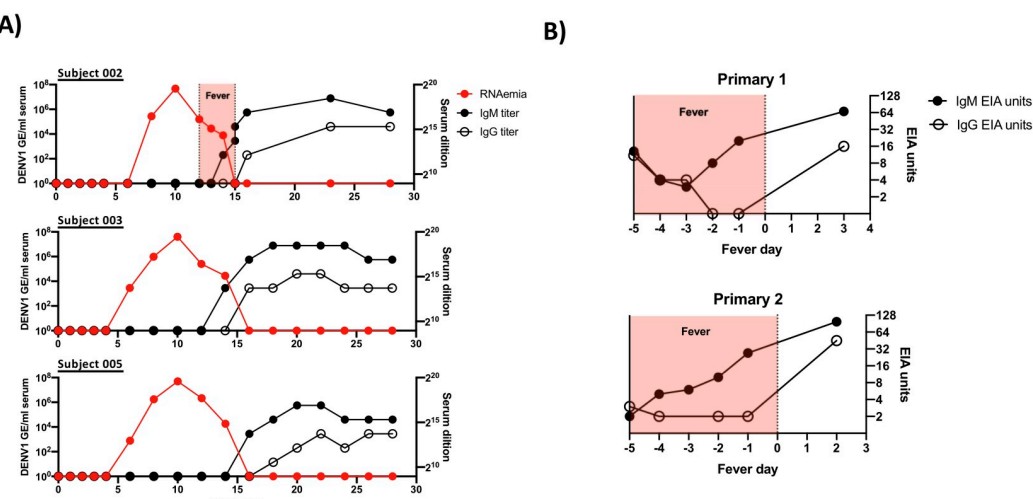

**Fig 1. Subject characterization and experimental overview. A)** Kinetics of RNAemia and DENV IgM/IgG seroconversion in experimental primary DENV-1 infection subjects. Serum IgM/IgG titers calculated as highest serum dilution providing a 2X signal over background. **B)** Kinetics of IgM/IgG seroconversion and duration of fever in natural primary DENV-1 infection subjects analyzed in this study. DENV-reactive serum IgM/IgG levels shown as EIA units.

## Leukocyte population identification and quantification

In order to assess the transcriptional signature associated with experimental and natural primary DENV-1 infection, high-throughput scRNAseq analysis was performed using the 10x Genomics 5' capture gene expression platform. Samples were sequenced to achieve a mean depth of 108,000 reads per cell, with an average of 5,707 cells captured per library (**S2 Table**). This final dataset contains a total of 171,208 high quality cells and 22 statistically distinct populations corresponding to all major anticipated leukocyte subsets (**Fig 2A and 2B** and **S3** and **S4** Tables). With the exception of neutrophils, all annotated populations were consistently observed in all experimental and natural primary DENV-1 infection samples included in this analysis (**Fig 2C** and **S3** and **S4** Tables). The relative distribution of all major leukocyte populations was consistent within each subject across all analyzed time point, with the notable exception of activated T cells and plasmablast phenotype B cells, both of which expanded in response to infection in the experimental primary DENV-1 infection subjects on study days 14/15 (**Figs 2D, S4, S5** and **S6**). No T cell or B cell activation/expansion was observed in the natural primary DENV-1 infection samples, constant with the time points analyzed [17]. To validate the population annotations as defined by scRNAseq, conventional flow cytometry was performed in parallel on the same samples and the frequencies of major leukocyte populations assessed (**S2** and **S3** Figs). The frequencies of all major leukocyte populations captured by either scRNAseq or flow cytometry exhibited a high degree of correlation across all time points in the experimental primary DENV-1 infection sample set (**S7 Fig**). No cell-associated DENV-1 RNA was observed in any sample at any time point following either experimental or natural primary DENV-1 infection.

## Identification of conserved experimental primary DENV-1 infection associated gene signatures

To determine the kinetics and composition of the subject/population-specific transcriptional response to experimental primary DENV-1 infection, we performed differentially expressed gene (DEG) analysis across all experimental primary DENV-1 infection samples. For this analysis, each annotated cell population on post-challenge study days was compared to the same population from the subject's baseline sample (study day 0) using a Wilcoxon Log Rank test with a Bonferroni correction for multiple comparisons. Only genes with corrected p value of < 0.05 also exhibiting a >0.5 fold change in expression were considered significant. Although some modest and inconsistent transcriptional changes were observed on study day 6 and 8, it was at study day 10 that a consistent and robust transcriptional response to experimental primary DENV-1 infection was observed across all subjects (**Fig 3A**). The cellular populations exhibiting the most dramatic response to experimental primary DENV-1 infection were conventional and CD16$^{hi}$ monocytes, with both populations containing > 100 DEGs on study day 10.

To better define the conserved "core" transcriptional signature associated with experimental primary DENV-1 infection, we reduced our differential transcriptional analysis to only contain genes that were differently expressed from baseline in all three experimental primary DENV-1 infection subjects at the indicated time point (**Fig 3B** and **S5 Table**). These conserved DEGs (cDEGs) were additionally separated into those induced by infection, and those suppressed by infection. This modified analysis again demonstrated that both conventional and CD16$^{hi}$ monocytes exhibited the most robust conserved transcriptional response to experimental primary DENV-1 infection, followed in magnitude by NK/NKT cells, and CD8$^{+}$ Effector Memory (E/M) cells (**Fig 3B** and **S5 Table**). The majority of the annotated cDEGs were upregulated by experimental primary DENV-1 infection, with only a minor population of

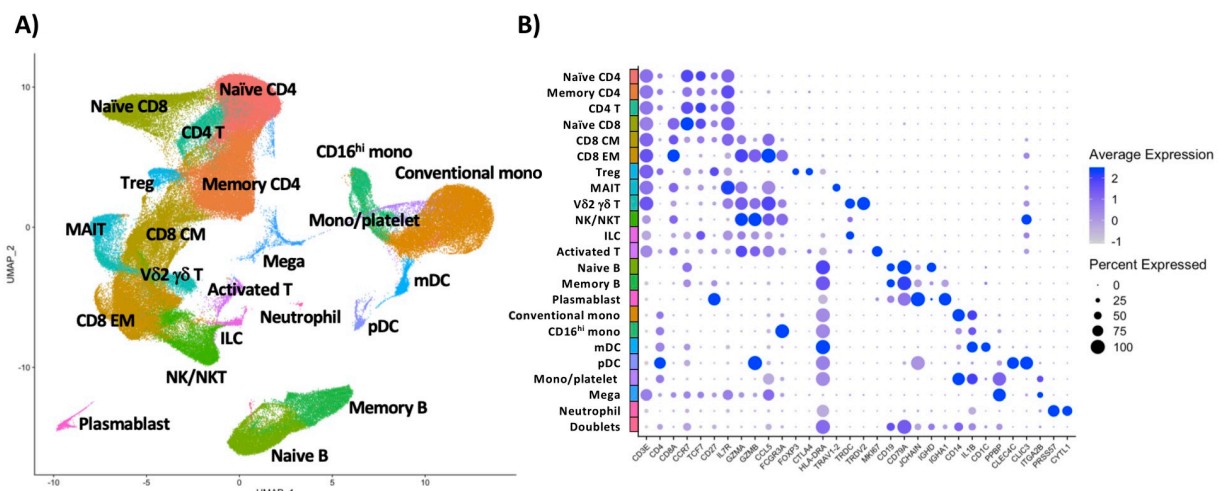

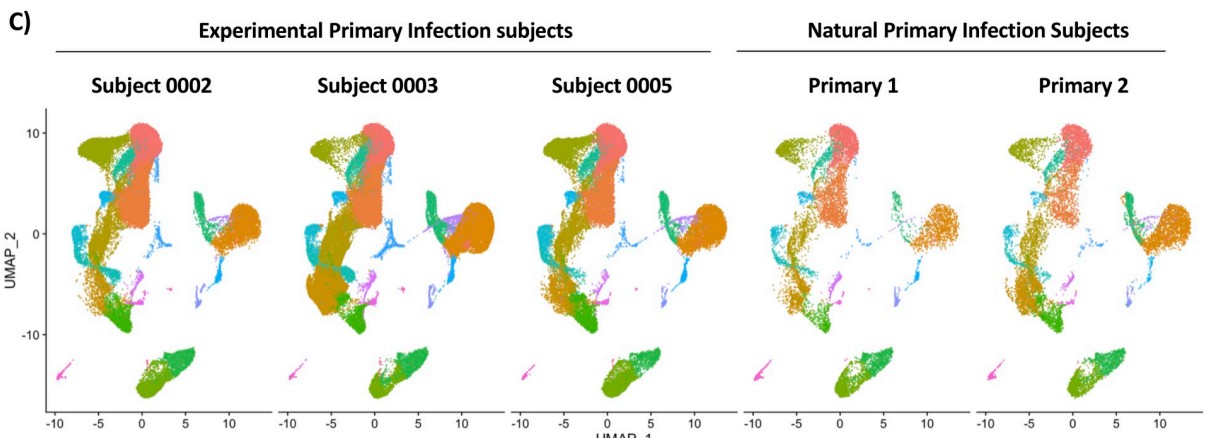

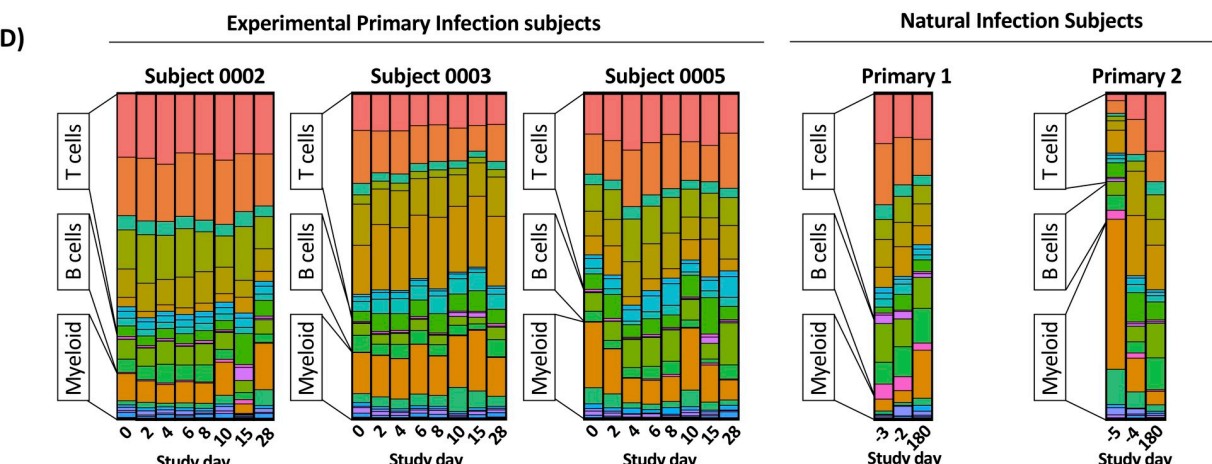

**Fig 2. Annotation and quantification of major leukocyte populations following experimental or natural primary DENV-1 infection. A)**
Integrated UMAP projection of scRNAseq data from all annotated leukocyte populations derived from all experimental primary DENV-1
infection subjects (n = 3, 8 time points per subject) and natural primary DENV-1 infection subjects (n = 2, 3 time points per subject). **B)**
Expression of key linage defining gene products across all annotated leukocyte populations captured in this analysis. **C)** Integrated UMAP
projection of all annotated leukocyte populations split by subject, including all time points. **D)** Relative population abundance in all subjects split
by time point.

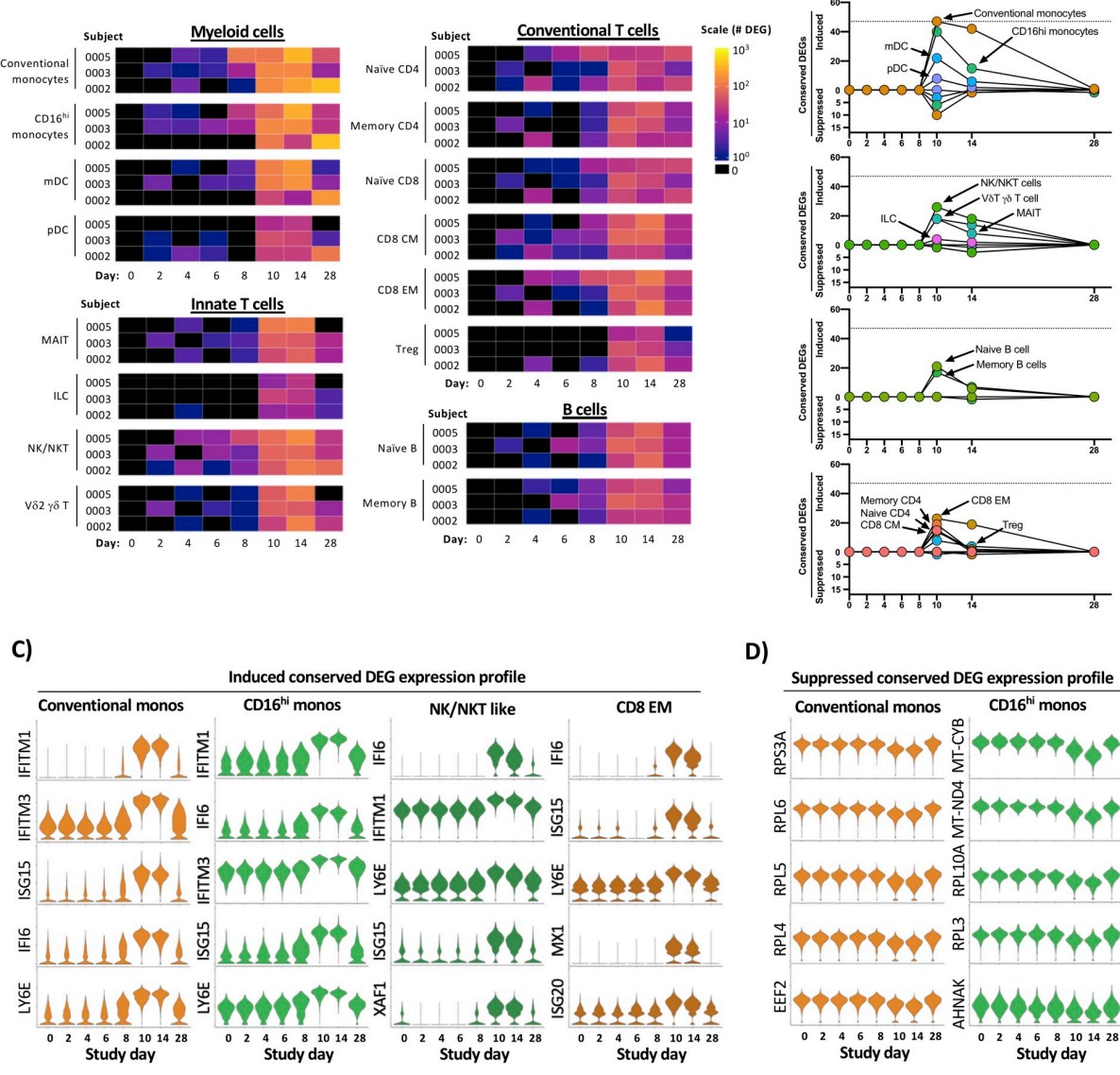

**Fig 3. Temporal and transcriptional characterization of experimental primary DENV-1 infection and identification of conserved gene signatures. A)** Quantification of differential gene expression across all major leukocyte populations following experimental primary DENV-1 infection. Subject- and population-specific differentially expressed genes (DEGs) were defined by a Wilcoxon rank-sum test with a Bonferroni correction relative to study day 0 for each subject. **B)** Population-restricted frequency and temporal dynamics of cDEGs across all experimental primary DENV-1 infection subjects. Conserved DEGs defined as DEGs observed in all three subjects at the same time point relative to baseline. **C)** Expression of select upregulated cDEGs from study day 10 within the indicated cell populations across all study time points. **D)** Expression of select suppressed cDEGs from study day 10 within the indicated cell populations across all study time points.

cDEGs consistently suppressed in response to experimental primary DENV-1 infection (**Fig 3B** and **S5 Table**). The genes that were most consistently and significantly upregulated following experimental primary DENV-1 infection primarily corresponded to interferon-induced gene products (e.g. IFITM1, IFI6, ISG15, and MX1) and other genes associated with acute inflammation (e.g. TRIM22 and LY6E) (**S5 Table**). The expression of these gene products consistently peaked on study days 10–14, and trended towards baseline by study day 28 (**Fig 3C** and **S5 Table**). The few cDEGs that were repressed in response to experimental primary

DENV-1 infection primarily corresponded to ribosomal protein subunits (e.g. RPL4, RPL5 and RPL6) translation elongation products (e.g. EEF2, EIF3L and EIF4B) mitochondrial associated gene products (e.g. MT-CYB and MT-ND4) (**Fig 3D and S5 Table**). This transcriptional profile is consistent with canonical IFN-associated suppression of protein translation and mitochondrial biogenesis and highlights a key mechanistic response to acute viral infection [18,19]. These conserved transcriptional responses to experimental primary DENV-1 infection not only reflect the acute cytokine-driven response to inflammation and infection but also capture the physiological response to inflammation and the induction of a distributed classic anti-viral cellular state in conventional and CD16hi monocytes.

## Identification of conserved natural primary DENV infection gene signatures

Having defined the kinetics and core conserved transcriptional profile associated with experimental primary DENV-1 infection, we performed the same analysis on the cells from individuals experiencing a natural primary DENV-1 infection. For this analysis, each annotated cell population from a given subject's acute infection sample set (acute #1 and acute #2) was compared to the same population from the subject's 6-month reference sample (**Fig 4A**). As was observed following experimental primary DENV-1 infection, the cellular populations exhibiting the most dramatic response to natural primary DENV-1 infection were conventional and CD16hi monocytes, with both populations containing > 300 DEGs in both acute time points in both subjects. However, the number of population-specific DEGs observed in response to natural primary DENV-1 infection was several fold higher than the response observed in the same population following experimental primary DENV-1 infection (**S8 Fig**).

To reduce the complexity of the data, we again endeavored to define a core set of population-specific cDEGs within the natural primary DENV-1 infection dataset. As the samples obtained from natural primary DENV-1 infection were collected on consecutive days and the exact timing post-infection is unknown, cDEGs for these samples were defined as those gene products differentially expressed in all acute infection time points relative to their respective reference sample (the intersection of the acute timepoint DEGs). Conventional monocytes, memory CD4+ T cells, and CD8+ EM cells exhibited the largest number of upregulated cDEG across all annotated cell populations, with the majority of the conserved upregulated DEGs corresponding to classic interferon-induced gene products (e.g. IFIT1, IFI44L, and ISG20) and other genes associated with acute inflammation (e.g. TRIM22 and LY6E) (**Fig 4B and 4C and S6 Table**). In contrast to the modest number of cDEGs suppressed in response to experimental primary DENV-1 infection, a significant fraction of the cDEGs identified in natural primary DENV-1 infection were suppressed relative to baseline (**Fig 4B**). However, similarly to those suppressed conserved DEGs identified following experimental primary DENV-1 infection, the suppressed cDEGs identified following natural primary DENV-1 infection primarily corresponded to ribosomal protein subunits (e.g. RPL4, RPL5, and RPL6), translation elongation products (e.g. EEF2, EIF3L, and EIF4B) and mitochondrial associated gene products (e.g. MT-CYB, and MT-ND4) (**Fig 4D and S6 Table**). These data suggest that natural primary DENV-1 infection not only induces the expression of interferon-induced gene products in a wide range of cell type, but that the physiological consequence this response is profound and widely distributed.

## Transcriptional profile overlap between experimental and natural primary DENV infections

Having identified cell-specific conserved sets of differently expressed genes present in PBMC subpopulations following experimental and natural primary DENV infection, we assessed the

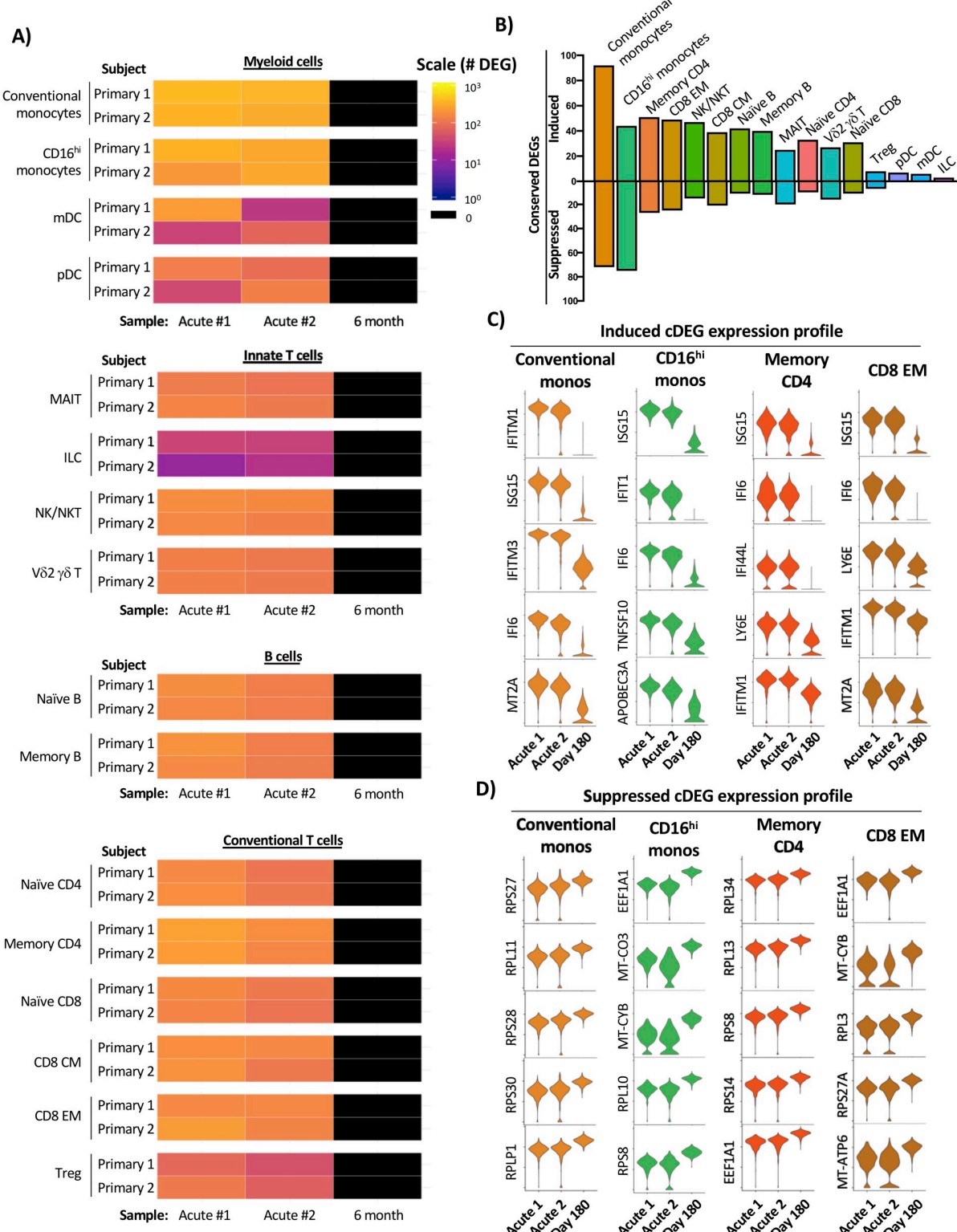

**Fig 4. Temporal and transcriptional characterization of natural primary DENV-1 infection and identification of conserved gene signatures. A)** Quantification of differential gene expression across all major leukocyte populations following natural primary DENV-1 infection. Subject- and population-specific differentially expressed genes (DEGs) were defined by a Wilcoxon rank-sum test with a Bonferroni correction relative to the 6-month control sample for each subject. **B)** Population-restricted frequency of cDEGs from both natural primary DENV-1 infection subjects. **C)** Expression of select induced cDEGs within the indicated cell populations across all study time points. **D)** Expression of select induced cDEGs within the indicated cell populations across all study time points.

overlap between these two datasets to compare the conserved transcriptional response to experimental and natural primary DENV infection. In light of the fact that cells obtained 10 days post infection in the experimental primary DENV-1 infection study contained the most cDEGs relative to baseline–and corresponded to the peak of RNAemia in all subjects—this time point was selected for comparison against the natural primary infection dataset.

Although the overall number of infection-induced cDEGs identified in cells from natural primary DENV-1 infection was greater than that observed following experimental primary DENV-1 infection, the cell population specific frequencies of infection-induced cDEGs in each population were highly correlated between the two study arms ($R^2 = 0.6825$, $p < 0.0001$) (**Fig 5A**). Furthermore, the infection-induced cDEGs identified following experimental primary DENV-1 were determined to consistently represent a subset of the cDEGs induced in natural primary DENV-1 infection (**Fig 5B** and **Table 1**). Gene ontogeny analysis revealed that the infection-induced cDEGs that were found in common to experimental and natural primary DENV infection datasets fell into classic interferon response pathways and negative regulation of viral replication (**Table 2**). Those cDEGs which were unique to natural primary DENV-1 infection corresponded to gene pathways involved in cytokine secretion, antigen processing/presentation, and detection of viruses (**Table 2**).

In contrast to the high degree of correlation observed in the cell-population specific frequencies of infection-induced cDEGs, infection-suppressed cDEGs were overwhelming restricted to those cells obtained following natural primary DENV-1 infection (**Fig 5C**). However, the few infection-suppressed cDEGs observed in cells following experimental primary DENV-1 infection generally represented a subset of those suppressed following natural primary DENV-1 infection (**Fig 5D** and **Table 3**). Infection-suppressed cDEGs overwhelming represented gene products associated with protein translation/elongation, as well as mitochondrial function and biogenesis (**Tables 3 and S6**). These data suggest that both natural and experimental primary DENV-1 infection upregulate a similar acute transcriptional program in similar cells, but that the greater magnitude of the that response elicited by natural primary DENV-1 infections has a significantly more profound physiological response on basic cellular processes, broadly suppressing protein translation and mitochondrial function. Furthermore, these data indicate that monocytes are uniquely responsive to both experimental and natural primary DENV infection, most notably in the suppression of gene products associated with protein translation and metabolic activity (**Fig 5E**).

## Discussion

In this study, we utilized high-throughput single-cell RNA sequencing (scRNAseq) technology to assess the longitudinal transcriptional profile following both experimental and natural primary DENV-1 infections with single-cell resolution. Core sets of conserved Differentially Expressed Genes that were induced or suppressed by either natural or experimental primary DENV-1 were identified, and the overlap between the study groups assessed. The infection-induced cDEGs associated with experimental primary DENV-1 infection were found to reflect a subset within the larger gene set associated with natural primary DENV-1 infection, primarily corresponding to gene products associated with a cellular response to systemic inflammation and interferon production. In contrast, infection-suppressed cDEGs were much more common in cells obtained from natural primary DENV-1 infection than in cells obtained from experimental primary DENV-1 infection. Infection-suppressed cDEGs primarily corresponded to gene products associated with protein translation/elongation and mitochondrial function: two cellular processes known to be suppressed by IFN signaling. These results are consistent with the notion that the immune response elicited by an experimental primary

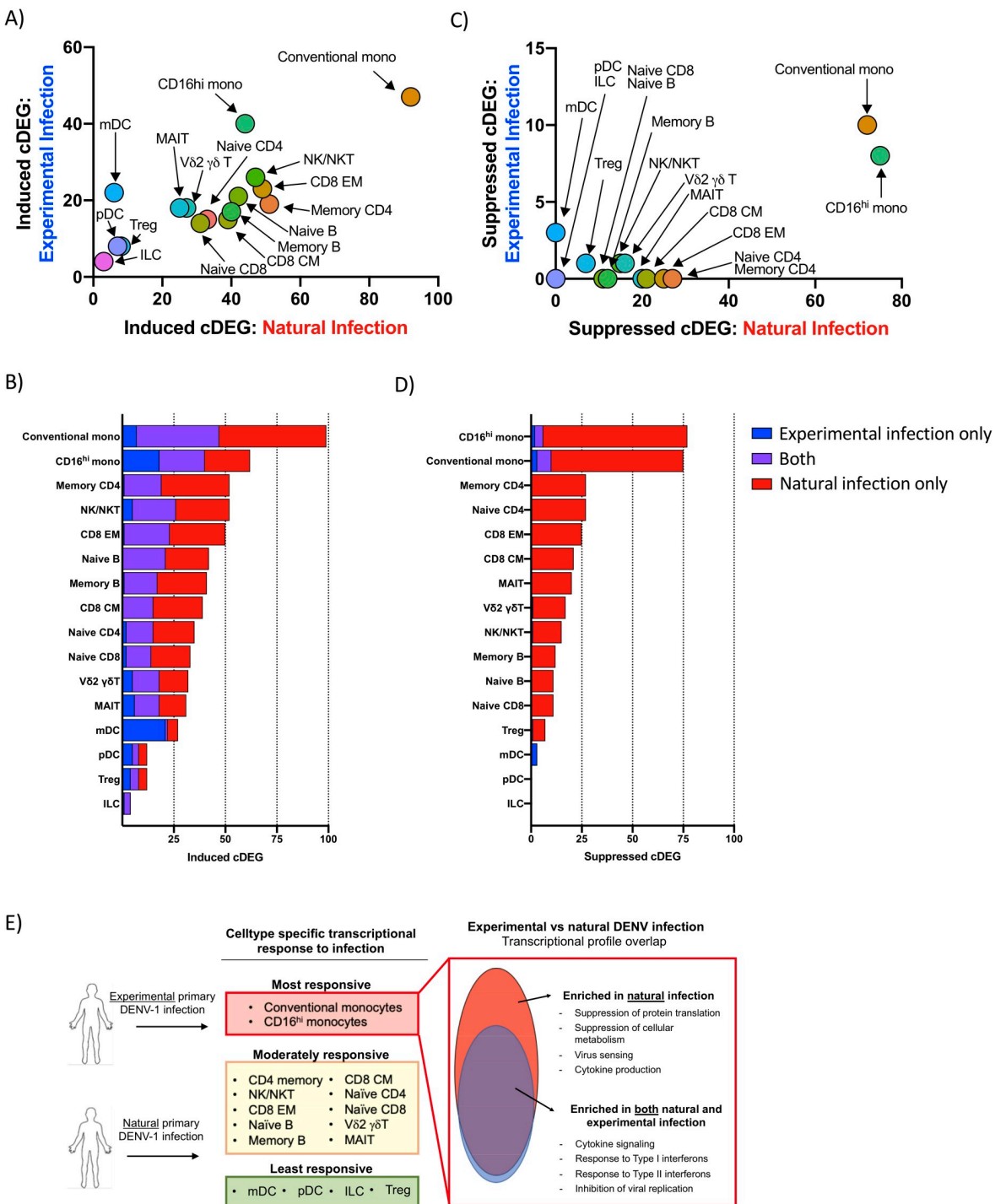

**Fig 5. Transcriptional profile overlap of experimental and natural primary DENV-1 infection A)** Frequency and cell-population distribution of infection-induced cDEGs identified following experimental or natural primary DENV-1 infection. Experimental primary DENV-1 infection cDEGs restricted to study day 10. **B)** Frequency and overlap of infection-induced cDEGs identified following experimental or natural primary DENV-1 infection **C)** Frequency and cell-population distribution of infection-suppressed cDEGs identified following natural primary or experimental DENV-1 infection. Experimental primary DENV-1 infection cDEGs restricted to study day 10. **D)** Frequency and overlap of infection-suppressed cDEGs identified following natural or experimental primary DENV-1 infection. **F)** Schematic representation of the differential gene expression responses observed following either natural or experimental primary DENV infection.

**Table 1. Conserved and unique infection-induced cDEG identified following natural primary DENV-1 infection or experimental primary DENV-1 infection on study day 10.**

| Population | Unique: Experimental primary DENV-1 | Common: Experimental AND natural primary DENV-1 infection | Unique: Natural primary DENV-1 infection |
|---|---|---|---|
| **Conventional monocytes** | HBB, HES4, NCF1, STAT2, TNFSF13B, TXNIP, WARS | APOBEC3A, EIF2AK2, EPSTI1, GBP1, IFI35, IFI44, IFI44L, IFI6, IFITM1, IFITM2, IFITM3, IRF7, ISG15, ISG20, LAP3, LY6E, MT2A, MX1, MX2, OAS1, OAS3, OASL, PARP14, PLAC8, PLSCR1, PSMB9, PSME2, RNF213, RSAD2, SAMD4A, SERPING1, SIGLEC1, STAT1, TMEM123, TNFSF10, TRIM22, TYMP, UBE2L6, VAMP5, XAF1 | BST2, CCL2, CCR1, CMPK2, CTSL, CXCL10, DDX58, DDX60L, DRAP1, FCGR1A, GBP4, GBP5, GCH1, GIMAP4, HERC5, HLA-A, HLA-B, HLA-C, HSH2D, IFI16, IFI27, IFIH1, IFIT1, IFIT2, IFIT3, IL1RN, LGALS3BP, LGALS9, MAFB, MARCKS, MYL12A, NAPA, NMI, NT5C3A, OAS2, PARP12, PARP9, PHF11, PML, PSMA4, RNASE2, SAMD9, SAMD9L, SCO2, SELL, SMCHD1, SP110, SPATS2L, TCN2, USP18, XRN1, ZBP1 |
| **CD16hi monocytes** | BST2, GBP1, HES4, IFITM1, IFITM3, LAP3, LY6E, OAS1, PARP14, PLAC8, PSMB9, PSME2, TMEM123, TNFSF13B, TYMP, UBE2L6, VAMP5, WARS | APOBEC3A, CXCL10, EIF2AK2, EPSTI1, HERC5, IFI35, IFI44, IFI44L, IFI6, IFIT2, IRF7, ISG15, MX1, MX2, NCF1, OAS3, PLSCR1, RSAD2, SERPING1, TNFSF10, TRIM22, XAF1 | CD300E, CDKN1A, DDX58, DDX60L, FFAR2, GBP4, GLUL, HLA-A, IFI27, IFIH1, IFIT1, IFIT3, LGALS3BP, LGALS9, MARCKS, OASL, SAMD9L, SIGLEC1, SPATS2L, TCN2, USP18, ZBP1 |
| **CD4 memory** | IFITM2 | BST2, EIF2AK2, EPSTI1, IFI44, IFI44L, IFI6, IFITM1, ISG15, ISG20, LY6E, MX2, PARP9, PLSCR1, PSMB9, SP100, STAT1, TRIM22, XAF1 | ADAR, DRAP1, GBP1, HERC5, HLA-A, HLA-B, HLA-E, IER2, IFI16, IFI35, IFIT3, IFITM3, IRF7, LGALS9, MT2A, MX1, OAS1, OAS3, OASL, PARP10, PARP12, PHF11, PSME2, RNF213, RSAD2, SAMD9, SAMD9L, SMCHD1, SP110, TNFSF10, TYMP, UBE2L6, ZBP1 |
| **NK/NKT** | IFITM2, LGALS1, PSMB9, PSME2, SHISA5 | BST2, EIF2AK2, EPSTI1, IFI35, IFI44L, IFI6, IFITM1, IFITM3, IRF7, ISG15, ISG20, LY6E, MX1, MX2, PARP9, PLSCR1, STAT1, TRIM22, TYMP, UBE2L6, XAF1 | ADAR, CD38, DTX3L, GBP1, IFI16, IFI44, IFIT1, IFIT3, LAG3, LGALS9, MT2A, NT5C3A, OAS1, OAS2, OAS3, OASL, PARP12, PRF1, RNF213, RSAD2, S100A11, SAMD9, SAMD9L, SP100, WARS, ZBP1 |
| **CD8 EM** | IFITM2 | BST2, DRAP1, EIF2AK2, IFI35, IFI44L, IFI6, IFITM1, IFITM3, IRF7, ISG15, ISG20, LY6E, MT2A, MX1, MX2, PLSCR1, PSMB9, PSME2, SP100, STAT1, TRIM22, XAF1 | ADAR, CD38, EPSTI1, GBP1, HERC5, IFI16, IFI44, IFIT1, IFIT3, LAG3, LGALS9, OAS1, OAS2, OASL, PARP10, PARP9, PRF1, RNF213, RSAD2, S100A11, SAMD9, SAMD9L, SP110, TAP1, TYMP, UBE2L6 |

DENV-1 infection represents a tempered version of that generated in response to a natural primary DENV-1 infection, but that the more pronounced inflammation associated with natural primary DENV-1 infection has a correspondingly distinct impact on basic cellular processes to induce a multi-layered systemic anti-viral state. These data provide insight into the molecular level response to primary DENV-1 infection, and how viral pathogenesis correlates with immune activation and cellular pathophysiology.

Interferon-induced gene products render cells inhospitable to viral replication and propagation through multiple mechanisms. These includes the expression of gene products that directly degrade viral RNA/DNA, the post-translational inhibition of viral entry and/or budding, post-translation inhibition of protein translation, as well as restriction of key metabolites and macromolecules required for virus replication and virion assembly [20]. Transcriptionally repressing the expression of gene products necessary for protein translation/elongation and mitochondrial function is one of the less specific–though highly effective–methods by which IFN exposure can limit viral replication [18,20]. This process of broadly inhibiting a function so central to cellular survival is understandably tightly regulated, and subject to multiple homeostatic feedback loops. While the data presented herein cannot formally demonstrate a causal relationship between the expression of IFN-induced gene products and the suppression of protein translational and cellular metabolism, it does suggests that a tipping point may exist,

**Table 2. Enriched GO terms in overlapping and natural infection only cell populations.**

| Cell type | Rank | GO term | Description | FE | FDR | GO term | Description | FE | FDR |
|---|---|---|---|---|---|---|---|---|---|
| | | Experimental AND Natural primary DENV-1 infection | | | | Natural primary DENV-1 infection ONLY | | | |
| Conventional mono | 1 | 0032020 | ISG15-protein conjugation | > 100 | 3.20E-02 | 0034344 | regulation of type III interferon production | > 100 | 1.30E-02 |
| Conventional mono | 2 | 0060700 | regulation of ribonuclease activity | > 100 | 5.92E-04 | 0035549 | positive regulation of interferon-beta secretion | > 100 | 2.42E-02 |
| Conventional mono | 3 | 0016185 | synaptic vesicle budding from presynaptic endocytic zone membrane | > 100 | 3.95E-02 | 0009597 | detection of virus | > 100 | 3.06E-02 |
| Conventional mono | 4 | 0035455 | response to interferon-alpha | > 100 | 1.15E-08 | 0035547 | regulation of interferon-beta secretion | > 100 | 3.03E-02 |
| Conventional mono | 5 | 0060337 | type I interferon signaling pathway | > 100 | 6.28E-25 | 0039528 | cytoplasmic pattern recognition receptor signaling pathway in response to virus | > 100 | 3.00E-02 |
| Conventional mono | 6 | 0071357 | cellular response to type I interferon | > 100 | 4.19E-25 | 0002480 | antigen processing and presentation of exogenous peptide antigen via MHC class I, TAP-independent | > 100 | 1.29E-03 |
| Conventional mono | 7 | 0034340 | response to type I interferon | > 100 | 8.74E-25 | 0002486 | antigen processing and presentation of endogenous peptide antigen via MHC class I via ER pathway, TAP-independent | 94.78 | 2.35E-03 |
| Conventional mono | 8 | 0045071 | negative regulation of viral genome replication | > 100 | 2.03E-19 | 0002484 | antigen processing and presentation of endogenous peptide antigen via MHC class I via ER pathway | 94.78 | 2.30E-03 |
| CD16$^{hi}$ monos | 1 | 0032020 | ISG15-protein conjugation | > 100 | 1.27E-02 | 0034344 | regulation of type III interferon production | > 100 | 6.83E-03 |
| CD16$^{hi}$ monos | 2 | 0016185 | synaptic vesicle budding from presynaptic endocytic zone membrane | > 100 | 1.59E-02 | 0035549 | positive regulation of interferon-beta secretion | > 100 | 1.11E-02 |
| CD16$^{hi}$ monos | 3 | 0140239 | postsynaptic endocytosis | > 100 | 2.31E-02 | 0009597 | detection of virus | > 100 | 1.29E-02 |
| CD16$^{hi}$ monos | 4 | 0098884 | postsynaptic neurotransmitter receptor internalization | > 100 | 2.25E-02 | 0035547 | regulation of interferon-beta secretion | > 100 | 1.26E-02 |
| CD16$^{hi}$ monos | 5 | 0070142 | synaptic vesicle budding | > 100 | 2.46E-02 | 0039528 | cytoplasmic pattern recognition receptor signaling pathway in response to virus | > 100 | 1.23E-02 |
| CD16$^{hi}$ monos | 6 | 0060337 | type I interferon signaling pathway | > 100 | 3.16E-16 | 1902741 | positive regulation of interferon-alpha secretion | > 100 | 2.56E-02 |
| CD16$^{hi}$ monos | 7 | 0071357 | cellular response to type I interferon | > 100 | 2.81E-16 | 1902739 | regulation of interferon-alpha secretion | > 100 | 2.51E-02 |
| CD16$^{hi}$ monos | 8 | 0099590 | neurotransmitter receptor internalization | > 100 | 4.18E-02 | 0071360 | cellular response to exogenous dsRNA | > 100 | 1.53E-03 |
| Memory CD4 | 1 | 0035455 | response to interferon-alpha | > 100 | 4.30E-06 | 0002519 | natural killer cell tolerance induction | > 100 | 5.10E-03 |
| Memory CD4 | 2 | 0035455 | response to interferon-beta | > 100 | 7.09E-08 | 0042270 | protection from natural killer cell mediated cytotoxicity | > 100 | 9.10E-03 |
| Memory CD4 | 3 | 0060337 | type I interferon signaling pathway | > 100 | 1.73E-14 | 0002480 | antigen processing and presentation of exogenous peptide antigen via MHC class I, TAP-independent | > 100 | 3.11E-04 |
| Memory CD4 | 4 | 0071357 | cellular response to type I interferon | > 100 | 1.30E-14 | 0060700 | regulation of ribonuclease activity | > 100 | 3.04E-04 |
| Memory CD4 | 5 | 0034340 | response to type I interferon | > 100 | 1.90E-14 | 0032020 | ISG15-protein conjugation | > 100 | 1.17E-02 |
| Memory CD4 | 6 | 0045071 | negative regulation of viral genome replication | > 100 | 1.24E-08 | 0002486 | antigen processing and presentation of endogenous peptide antigen via MHC class I via ER pathway, TAP-independent | > 100 | 5.99E-04 |
| Memory CD4 | 7 | 0048525 | negative regulation of viral process | 89.11 | 3.23E-11 | 0002484 | antigen processing and presentation of endogenous peptide antigen via MHC class I via ER pathway | > 100 | 5.87E-04 |
| Memory CD4 | 8 | 1903901 | negative regulation of viral life cycle | 81.77 | 7.16E-08 | 2001187 | positive regulation of CD8-positive, alpha-beta T cell activation | > 100 | 1.71E-02 |

(*Continued*)

**Table 2.** (Continued)

| Cell type | Rank | Experimental AND Natural primary DENV-1 infection | | | | Natural primary DENV-1 infection ONLY | | | |
|---|---|---|---|---|---|---|---|---|---|
| | | GO term | Description | FE | FDR | GO term | Description | FE | FDR |
| NK/NKT | 1 | 0016185 | synaptic vesicle budding from presynaptic endocytic zone membrane | > 100 | 1.02E-02 | 0060700 | regulation of ribonuclease activity | > 100 | 8.34E-07 |
| NK/NKT | 2 | 0140239 | postsynaptic endocytosis | > 100 | 1.28E-02 | 0032069 | regulation of nuclease activity | > 100 | 1.62E-05 |
| NK/NKT | 3 | 0098884 | postsynaptic neurotransmitter receptor internalization | > 100 | 1.91E-02 | 0060337 | type I interferon signaling pathway | > 100 | 1.22E-12 |
| NK/NKT | 4 | 0035455 | response to interferon-alpha | > 100 | 1.87E-02 | 0071357 | cellular response to type I interferon | > 100 | 9.73E-13 |
| NK/NKT | 5 | 0070142 | synaptic vesicle budding | > 100 | 3.96E-08 | 0035455 | response to interferon-alpha | 96.53 | 2.38E-03 |
| NK/NKT | 6 | 0035456 | response to interferon-beta | > 100 | 2.14E-02 | 0034340 | response to type I interferon | 96.53 | 1.48E-12 |
| NK/NKT | 7 | 0060337 | type I interferon signaling pathway | > 100 | 8.49E-10 | 0045071 | negative regulation of viral genome replication | 87.19 | 2.11E-09 |
| NK/NKT | 8 | 0071357 | cellular response to type I interferon | > 100 | 8.20E-21 | 0060333 | interferon-gamma-mediated signaling pathway | 75.08 | 5.36E-09 |
| CD8 EM | 1 | 0016185 | synaptic vesicle budding from presynaptic endocytic zone membrane | > 100 | 1.21E-02 | 0060700 | regulation of ribonuclease activity | > 100 | 1.65E-04 |
| CD8 EM | 2 | 0140239 | postsynaptic endocytosis | > 100 | 1.77E-02 | 0016185 | synaptic vesicle budding from presynaptic endocytic zone membrane | > 100 | 1.54E-02 |
| CD8 EM | 3 | 0098884 | postsynaptic neurotransmitter receptor internalization | > 100 | 1.74E-02 | 0140239 | postsynaptic endocytosis | > 100 | 2.26E-02 |
| CD8 EM | 4 | 0035455 | response to interferon-alpha | > 100 | 4.36E-08 | 0098884 | postsynaptic neurotransmitter receptor internalization | > 100 | 2.22E-02 |
| CD8 EM | 5 | 0070142 | synaptic vesicle budding | > 100 | 1.94E-02 | 0070142 | synaptic vesicle buddin | > 100 | 2.61E-02 |
| CD8 EM | 6 | 0060337 | type I interferon signaling pathway | > 100 | 1.06E-22 | 0035455 | response to interferon-alpha | > 100 | 1.76E-05 |
| CD8 EM | 7 | 0071357 | cellular response to type I interferon | > 100 | 5.32E-23 | 0060340 | positive regulation of type I interferon-mediated signaling pathway | > 100 | 3.93E-02 |
| CD8 EM | 8 | 0035456 | response to interferon-beta | > 100 | 9.32E-10 | 0035457 | cellular response to interferon-alpha | > 100 | 3.86E-02 |

**Table 3. Conserved and unique infection-suppressed cDEG identified following natural primary DENV-1 infection or experimental primary DENV-1 infection on study day 10.**

| Population | Unique: Experimental primary DENV-1 | Common: Experimental AND natural primary DENV-1 infection | Unique: Natural primary DENV-1 infection |
|---|---|---|---|
| CD16hi monocytes | IL1B, AHNAK | RPL10A, RPL3, MT-ND4, MT-CYB | RPL6, RPS3A, EEF1A1, RPS15A, RPL10, RPS13, RPS8, RPS23, RPL34, RPS4X, RPS27A, RPL8, RPL14, RPS14, RPS7, RPL18A, RPS6, RPL12, RPLP2, RPL35A, RPL18, RPL11, RPL21, RPL7A, RPL9, RPL5, SLC25A6, RPL32, RPS24, RPS21, RPL13, RPL30, RPL4, MT-CO3, RPL19, RPL22, RPS16, RPL26, RPL15, GNB2L1, COTL1, PABPC1, SLC25A5, ACTG1, EEF2, EEF1B2, NACA, RPLP0, RPL27, RPS25, RPS5, TPT1, RPL23A, RPS3, RPSA, RPS18, PFDN5, EIF3K, RPL37A, UQCRB, BTF3, RPL29, MT-ND3, RPS4Y1, RPL23, YBX1, CCNI, RPS17, NAP1L1, EIF3L, RBM3 |
| Conventional monocytes | IL1B, CXCL8, EIF4B | RPS3A, RPL4, EEF2, RPL6, RPL5, EIF3L, RPL3 | TPT1, RPS28, EEF1A1, RPS13, RPLP1, RPL8, RPL18, RPS7, RPS14, RPL34, RPS24, RPL35A, RPS15A, MT-ND3, MT-CO3, RPS8, RPS4X, RPL11, RPL10, RPL19, RPL9, NACA, RPL22, RPL26, RPL37, RPS23, RPL15, RPL7A, RPL32, RPL18A, BTF3, RPL30, RPLP2, RPS16, RPS27A, RPS6, RPLP0, RPL29, COX4I1, PABPC1, CSTA, COTL1, SLC25A6, RPL27, RPL13, RPL10A, RPS5, EEF1B2, RPL21, AP1S2, RPS3, GSTP1, RPS18, PCBP2, RPL7, GNB2L1, SLC25A5, RPS4Y1, ALDH2, EIF3F, RBM3, FCGRT, EIF3E, EIF3H, CRTAP |

where the magnitude of the interferon-induced transcriptional response reaches a threshold, whereupon a broader range of cellular factors are significantly impacted causing a qualitative change in transcriptional response to infection.

The observation that both the induction of IFN-responsive gene products as well as the suppression of genes associated with protein translation and mitochondrial function was most dramatically and consistently observed in conventional and CD16[hi] monocytes may help explain certain discrepancies in the literature regarding DENV cellular tropism. Historically, monocytes, macrophages, and dendritic cells have been cited as the primary circulating cellular reservoir of DENV [21,22]. However, this assertion was primarily based primarily on *in vitro* infection experiments, where the impact of systemic inflammation and interferon signaling may not impact cellular permissiveness to infection. More recent studies utilizing flow cytometry, RT-PCR, and scRNAseq analysis have suggested that B cells are the principal circulating natural reservoir of DENV [23–28]. Our observation that monocytes appear to respond more dramatically to DENV-elicited systemic inflammation than B cells suggests that while monocytes are more permissive to infection *in vitro*, they become a much less attractive target of infection and replication *in vivo*. However, while a profound transcriptional response to both experimental and natural primary DENV-1 infection was observed in this analysis, it is notable that we were unable to detect any cell-associated viral RNA from any subject or sample. This despite the fact that we have previously demonstrated that the scRNAseq platform used in this analysis is capable of identifying and quantifying cells harboring DENV genomic RNA [28]. These previous studies utilized either *in vitro* infected cells or PBMC from individuals experiencing a secondary DENV infection, which has been shown to result in a higher cell-associated DENV burden [23]. Therefore, the lower viral burden associated with primary DENV infection may not provide enough of a signal to be reliably captured using this analysis platform. It is also acknowledged here that DENV has demonstrated tropism for tissues such as skin, spleen, lymph nodes, liver and bone marrow, and hence, tissue-resident infected cells may be important drivers of systemic inflammation [29–31]. Consequently, the relative abundance of virally infected cells–and their role in driving the inflammatory profile described herein–remains unresolved.

Although the data presented herein represents an extremely high-resolution transcriptional assessment of both natural and experimental primary DENV-1 infection, several limitations of the study must be acknowledged. Firstly, the relatively small numbers of subjects analyzed in both arms of the study limits the power of the analysis. Secondly, the significant age and background differences between the subjects are potential confounders for the analysis. In addition to viral genetic differences, the route and method of virus inoculation differs significantly between natural and experimental primary DENV-1 infection (mosquito bite versus subcutaneous needle). The impact of these variables on the immunological profile described here can only be addressed with further focused studies guided by the data presented here.

Taken together, the data presented in this study demonstrates that the immunological response elicited by the administration of an attenuated DENV in an experimental setting involves many of the same molecular pathways in the same cells as wild-type, naturally-acquired primary DENV. This includes the activation and expansion of lymphocyte populations (T/NK/NKT cells and plasmablasts), as well as predictable perturbation of the transcriptional profile in all major leukocyte subsets. However, natural infection results in a more pronounced pattern of inflammatory gene upregulation accompanied by marked suppression of gene products associated with protein translation and mitochondrial function, particularly among monocyte populations. This study highlights the pivotal role that monocytes play in responding to acute DENV infection, potentially offering a framework to develop more accurate and rapid diagnostic metrics in the future.

## Materials and methods

### Ethics statement

The Dengue Human Challenge Model and associated analysis was approved by the State University of New York Upstate Medical University (SUNY-UMU) and the Department of Defense's Human Right Protection Organization (Identification: WRAIR #2237). The Natural primary DENV infection sample collection protocol and associated analysis was approved by the Institutional Review Boards of the Thai Ministry of Public Health, the Office of the U.S. Army Surgeon General, and the University of Massachusetts Medical School (Identification: WRAIR #1620).

### Dengue human challenge model

Peripheral blood mononuclear cells (PBMC) and plasma for this study were obtained from the previously described phase 1 open label, Dengue Virus-1 Live Virus Human Challenge (DENV-1-LVHC) study performed at the State University of New York, Upstate Medical University in Syracuse, NY [12]. The study was approved by the State University of New York Upstate Medical University (SUNY-UMU) and the Department of Defense's Human Right Protection Organization (HRPO). ClinicalTrials.gov identifier for this trial is NCT02372175. All subjects included in this study received $3.25 \times 10^3$ PFU of the 45AZ5 DENV-1 challenge strain virus following pre-screening to ensure an absence of preexisting DENV immunity. The 45AZ5 DENV-1 challenge strain used in this study was generated by serial passage of the parental Nauru/West Pac/1974 DENV-1 isolate through diploid fetal rhesus lung cell line (FRhL) in the presence of 5-azacytidine, followed by plaque cloning and secondary amplification in FRhL cells [15,16]. Flavivirus antibody screening, dengue IgM and IgG ELISA, micro-neutralization assay and quantitative reverse-transcriptase polymerase chain reaction (RT-PCR) were performed at the Viral Diseases Branch, Walter Reed Army Institute of Research (WRAIR) in Silver Spring, Maryland using previously published techniques [32–35].

### Natural primary DENV infection sample collection

PBMC and plasma were isolated from whole blood specimens obtained from children enrolled in a hospital-based acute febrile illness study in Bangkok, Thailand, the design of which has been previously described [36,37]. In brief, the study enrolled children who presented to the hospital with acute febrile illness. Blood samples were obtained daily during illness and at early and late convalescent time points; the term 'fever day' is used to report acute illness time points relative to Day 0, defined as the day of defervescence. The infecting virus type (DENV-1-4) was determined by RT-PCR and/or virus isolation as previously described [38], and serology (EIA and HAI assays) was used to distinguish primary and secondary DENV infections [39]. IgM/IgG EIA values were calculated using a standard binding index assay and the following formula: EIA units = ((OD (average test sample)–OD (average negative control)) /(OD (average weak positive control)–OD (average negative control)) * 100). Written informed consent was obtained from each subject and/or his/her parent or guardian. The study protocol was approved by the Institutional Review Boards of the Thai Ministry of Public Health, the Office of the U.S. Army Surgeon General, and the University of Massachusetts Medical School. PBMC and plasma samples were cryopreserved for later analysis.

### Flow cytometry

Cryopreserved PBMC were thawed and placed in RPMI 1640 medium supplemented with 10% heat-inactivated fetal bovine serum, L-glutamine, penicillin, and streptomycin prior to

analysis. Cell viability was assessed using CTL-LDC Dye (Cellular Technology Limited [CTL], Shaker Heights, OH) and a CTL-ImmunoSpot S6 Ultimate-V Analyzer (CTL). Surface staining for flow cytometry analysis was performed in PBS supplemented with 2% FBS at room temperature. Aqua Live/Dead or Violet Live/Dead dye (ThermoFisher) was used to exclude dead cells in all experiments. Antibodies and dilutions used for flow cytometry analysis are listed in **S7 Table.** Flow cytometry analysis was performed on a custom-order BD LSRFortessa instrument, and data analyzed using FlowJo v10.2 software (Treestar).

## Single-cell RNA sequencing library generation

Thawed PBMC suspensions were prepared for single-cell RNA sequencing using the Chromium Single-Cell 5′ Reagent version 2 kit and Chromium Single-Cell Controller (10x Genomics, CA)[40]. 2000–8000 cells per reaction suspended at a density of 50–500 cells/μL in PBS plus 0.5% FBS were loaded for gel bead-in-emulsion (GEM) generation and barcoding. Reverse transcription, RT-cleanup, and cDNA amplification were performed to isolate and amplify cDNA for downstream 5′ gene or enriched V(D)J library construction according to the manufacturer's protocol. Libraries were constructed using the Chromium Single-Cell 5′ reagent kit, V(D)J Human B Cell Enrichment Kit, 3′/5′ Library Construction Kit, and i7 Multiplex Kit (10x Genomics, CA) according to the manufacturer's protocol.

## Sequencing

scRNAseq 5′ gene expression libraries and BCR V(D)J enriched libraries were sequenced on an Illumina NovaSeq 6000 instrument using the S2, or S4 reagent kits (300 cycles). Libraries were balanced to allow for ~150,000 reads/cell for 5′ gene expression libraries. Sequencing parameters were set for 150 cycles for Read1, 8 cycles for Index1, and 150 cycles for Read2. Prior to sequencing, library quality and concentration were assessed using an Agilent 4200 TapeStation with High Sensitivity D5000 ScreenTape Assay and Qubit Fluorometer (Thermo Fisher Scientific) with dsDNA BR assay kit according to the manufacturer's recommendations.

## 5' gene expression analysis/visualization

5′ gene expression alignment from all PBMC samples was performed using the 10x Genomics Cell Ranger pipeline [40]. Sample demultiplexing, alignment, barcode/UMI filtering, and duplicate compression was performed using the Cell Ranger software package (10x Genomics, CA, v2.1.0) and bcl2fastq2 (Illumina, CA, v2.20) according to the manufacturer's recommendations, using the default settings and mkfastq/count commands, respectively. Transcript alignment was performed against a human reference library generated using the Cell Ranger mkref command, the Ensembl GRCh38 v87 top-level genome FASTA, and the corresponding Ensembl v87 gene GTF.

Multi-sample integration, data normalization, dimensional reduction, visualization, and differential gene expression were performed using the R package Seurat (v3.1.4) [41,42]. All datasets were filtered to only contain cells with between 200–6,000 unique features and <10% mitochondrial RNA content. To eliminate erythrocyte contamination, datasets were additionally filtered to contain cells with less than a 5% erythrocytic gene signature (defined as HBA1, HBA2, HBB). Data were scaled, normalized, and transformed prior to multi-sample integration using the negative binomial regression model of the Seurat SCTransform() function [43]. SelectIntegrationFeatures() and PrepSCTIntegration() functions were used to identify conserved features for dataset integration, and final dataset anchoring/integration were performed using FindIntegrationAnchors() and IntegrateData() functions, with the day 0 experimental DENV-1 infection samples and day 180 natural primary DENV-1 infection samples used as

reference datasets. PCA was performed using variable genes defined by SCTransform() additionally filtered to remove TCR V/D/J or BCR κ/λ gene segments. The first 33 resultant PCs were used to perform a UMAP dimensional reduction of the dataset (RunUMAP()) and to construct a shared nearest neighbor graph (SNN; FindNeighbors()). This SNN was used to cluster the dataset (FindClusters()) with default parameters and resolution set to 0.4.

Following dataset integration and dimensional reduction/clustering, gene expression data was $\log_e$(UMI+1) transformed and scaled by a factor of 10,000 using the NormalizeData() function. This normalized gene expression data was used to determine cellular cluster identity by utilizing the Seurat application of a Wilcoxon rank-sum test (FindAllMarkers()), and comparing the resulting differential expression data to known cell-linage specific gene sets. Differential gene expression analysis between study time points was performed using normalized gene expression data and the Wilcoxon rank-sum test with implementation in the FindMarkers() function, with a $\log_e$ fold chain threshold of 0.5 and min.pct of 0.25. Bonferroni correction was used to control for Fall Discovery Rate (FDR), with a corrected p value of $< 0.05$ considered significant.

## Statistical analysis

Differential gene expression analysis of scRNAseq data was performed using normalized gene expression counts and the Wilcoxon rank-sum test in the Seurat FindMarkers() function [41,42]. A $\log_e$ fold chain threshold for gene expression changes of 0.5 and min.pct of 0.25 was used for all comparisons, and a Bonferroni correction was used to control for Fall Discovery Rate (FDR). A corrected p value of $< 0.05$ considered significant in conjunction with the additional filters above. All other statistical analysis was performed using GraphPad Prism 8 Software (GraphPad Software, La Jolla, CA). A $P$-value $< 0.05$ was considered significant.

## Supporting information

**S1 Fig. Schematic representation of the analysis scheme utilized in this study.**
(TIF)

**S2 Fig. Flow cytometry gating scheme for monocyte, DC, and B cell phenotyping from experimental primary DENV-1 infection samples.**
(TIF)

**S3 Fig. Flow cytometry gating scheme for T and NK cell phenotyping for experimental primary DENV-1 infection samples.**
(TIF)

**S4 Fig.** Experimental primary DENV-1 associated lymphocyte expansion, A) Expansion of transcriptionally defined plasmablasts as a fraction of all B cells in response to experimental primary DENV-1 infection B) Expansion of transcriptionally defined activated T cells as a fraction of all T cells in response to experimental primary DENV-1 infection.
(TIF)

**S5 Fig. Plasmablast expansion following experimental primary DENV-1 infection.** Cells are gated on CD3⁻CD56⁻CD19⁺ viable lymphocytes.
(TIF)

**S6 Fig. CD4⁺ T cell and CD8⁺ T cell activation following experimental primary DENV-1 infection.** T cells defined as viable CD3⁺ CD56⁻ lymphocytes.
(TIF)

**S7 Fig. Correlation between population frequencies as defined by scRNAseq or flow cytometry in all experimental primary DENV-1 infection samples.** Population frequency defined as percent of all cells.
(TIF)

**S8 Fig. Number of differentially express genes in the indicated populations relative to baseline.** Experimental primary DENV-1 samples are from study day 10. Natural primary DENV-1 infection samples are from acute 1/acute 2 time points ** p <0.01, *** p <0.001, unpaired two-tailed t test.
(TIF)

**S1 Table. Natural primary DENV-1 infection subject information.**
(DOCX)

**S2 Table. Sample sequencing metrics.**
(DOCX)

**S3 Table. Sample/population frequency: T cell populations.**
(DOCX)

**S4 Table. Sample/population frequency: B cells and myeloid linage cells.**
(DOCX)

**S5 Table. Conserved differentially expressed genes: day 10 Experimental primary DENV-1 infection.**
(DOCX)

**S6 Table. Core differentially expressed genes: natural primary DENV-1 infection.**
(DOCX)

**S7 Table. Antibodies used for flow cytometry.**
(DOCX)

## Acknowledgments

### Disclaimer

The opinions or assertions contained herein are the private views of the authors and are not to be construed as reflecting the official views of the US Army or the US Department of Defense, or the National Institutes of Health. Material has been reviewed by the Walter Reed Army Institute of Research. There is no objection to its presentation and/or publication. The investigators have adhered to the policies for protection of human subjects as prescribed in AR 70–25.

## Author Contributions

**Conceptualization:** Adam T. Waickman, Richard G. Jarman, Alan L. Rothman, Timothy Endy, Jeffrey R. Currier.

**Data curation:** Adam T. Waickman.

**Formal analysis:** Adam T. Waickman, Michael K. McCracken.

**Funding acquisition:** Adam T. Waickman, Alan L. Rothman, Jeffrey R. Currier.

**Investigation:** Adam T. Waickman, Heather Friberg, Gregory D. Gromowski, Wiriya Rutvisuttinunt, Tao Li, Hayden Siegfried, Kaitlin Victor.

**Methodology:** Adam T. Waickman, Heather Friberg, Gregory D. Gromowski, Wiriya Rutvisuttinunt, Michael K. McCracken, Richard G. Jarman, Alan L. Rothman, Timothy Endy, Jeffrey R. Currier.

**Project administration:** Stefan Fernandez, Anon Srikiatkhachorn, Damon Ellison, Richard G. Jarman, Stephen J. Thomas, Alan L. Rothman, Timothy Endy, Jeffrey R. Currier.

**Supervision:** Heather Friberg, Gregory D. Gromowski, Wiriya Rutvisuttinunt, Stefan Fernandez, Anon Srikiatkhachorn, Damon Ellison, Richard G. Jarman, Stephen J. Thomas, Alan L. Rothman, Timothy Endy, Jeffrey R. Currier.

**Visualization:** Adam T. Waickman, Heather Friberg.

**Writing – original draft:** Adam T. Waickman, Jeffrey R. Currier.

**Writing – review & editing:** Adam T. Waickman, Heather Friberg, Jeffrey R. Currier.

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
