## [Decision Letter · Decision Letter 0]

22 Oct 2020

Dear Dr Waickman,

Thank you very much for submitting your manuscript "Temporally integrated single cell RNA sequencing analysis of controlled and natural primary human DENV-1 infections" for consideration at PLOS Pathogens. As with all papers reviewed by the journal, your manuscript was reviewed by members of the editorial board and by several independent reviewers. In light of the reviews (below this email), we would like to invite the resubmission of a significantly-revised version that takes into account the reviewers' comments.

We cannot make any decision about publication until we have seen the revised manuscript and your response to the reviewers' comments. Your revised manuscript is also likely to be sent to reviewers for further evaluation.

Sincerely,

Daniela Weiskopf

Guest Editor

PLOS Pathogens

Ana Fernandez-Sesma

Section Editor

PLOS Pathogens

Kasturi Haldar

Editor-in-Chief

PLOS Pathogens

orcid.org/0000-0001-5065-158X

Michael Malim

Editor-in-Chief

PLOS Pathogens

orcid.org/0000-0002-7699-2064

Reviewer's Responses to Questions

**Part I - Summary**

Reviewer #1: This is a very interesting paper that attempts to compare host RNAs by single cell RNA sequencing of cells derived from volunteers who received a candidate live attenuated dengue-1 vaccine used for controlled human infection models (DHIM) compared to those who were infected by wild-type dengue-1 virus (natural infection). I greatly admire the authors for undertaking such a difficult and complicated project.

The data are interesting but there are a number of questions regarding the exact situation studied that need to be addressed in order to adequately understand the data.

Technically, the authors downsampled the RNA data (which is important) but it is not explicitly stated that downsampling was to the same number of reads per cell in the analysis. This is critical.

Figure 1 provides information on the three subjects from the DHIM group but the information on the two children who had natural infection is in the supplementary data (supplementary Table 1 and supplementary figure 1). These data all need to be in figure 1. The authors need to explain why the IgG/IgM values are different between the two groups. It would be very helpful to show viremia data for the three timepoints of the two natural infection subjects. I say this as it is clear the best data for the DHIM group comes from the peak viremia samples (day 10), and viremias are very similar titers and this point is made on line 284 by the authors. However, the parameters of the samples from the two natural infections need to be stated, in particular viremia data on the timepoints examined. The authors do not discuss why these samples are the best in the Discussion. This is needed as lines 361-362 in the Discussion are very vague. The possibilities that the viremias for the natural infections are not equivalent to those in the DHIM samples needs to be considered.

Why use only three DHIM subjects and two with natural infections. How was selection of subjects made? Does the difference in ages of subjects in the two groups contribute to the differences in results?

The paper needs a summary figure to provide an overview for readers for a very complex paper. Currently, there is a supplementary figure 9. This needs to be moved to the main paper and expanded.

Presumably there are viral genome differences between the attenuated DENV-1 in the DHIM and the viruses in the natural infection. If possible, the sequence differences (NT and deduced AA) between these viruses needs to be included in the text to help understand the viral determinants for the differences between the two types of infection. Furthermore, it is disappointing that the authors have not speculated on the molecular basis of attenuation of the DHIM virus. This is a potentially important spinoff of the paper.

Lines 329-330 of the Discussion are vague. Clearly, there is “insights” from the results but these need to be spelt out.

Reviewer #2: This study provides information on the transcriptional response of peripheral blood cells during dengue infection at a single cell level and would be a valuable tool for mining the transcriptional profile for new hypotheses relating to dengue infection control and severity. However, there are some drawbacks to the study design. While I am sympathetic to the cost of the study and the uniqueness of the information provided by this sequencing technique, a significant drawback is the limited numbers of patients from which the data are derived, the mis-matched age comparisons and, as a consequence, the limited robustness of the statistical plan. Although limited in N of total patients, the numbers of time points of longitudinal analysis are an advantage of the study. Overall, the conclusions of the study might be more strong if the focus of the study shifted to the longitudinal changes in paired samples through the course of experimental infection, as emphasized by the title and nicely presented in Fig. 3. This could also be more clearly defined using the N available rather than comparisons between 2 and 3 patient samples per group for the natural versus controlled infection description where differences in IFN and protein translation pathways are reported. In short, this is a valuable data set but the authors should be more cautious which conclusions can be drawn from it and not all conclusions are well supported. There is also an incomplete description of the corresponding infection kinetics (for the natural infection group) which would allow better interpretation of the data.

Reviewer #3: Here, 3 experimental human infections with live-attenuated DENV-1 are compared with 2 cases of natural DENV-1 infections by scRNAseq. Very similar inflammatory differential gene regulation was observed. Natural infection was associated with a more pronounced suppression of translation and mitochondrial genes in monocytes, consistent with increased replication of the virus.

**Part II – Major Issues: Key Experiments Required for Acceptance**

Reviewer #1: No new experiments are needed but a lot of clarifications are needed to interpret the paper.

Reviewer #2: The terminology of “controlled” DENV infection see a little non-standard and I would suggest reserving for the discussion. I would suggest considering other alternatives to convey that this is not a fully virulent challenge model. This experimental DENV infection is with an attenuated vaccine candidate, right? Might this be the effect described be of virus attenuation only rather than the same gene patterns that regulate natural infection control of non-attenuated strains? I think this caveat needs to be discussed further and the abstract/intro should be more cautiously worded regarding what the experimental set up can establish.

One important distinction in the comparison is that samples from adults and children are compared. It would have been much stronger if the Thai natural infection samples had been case controlled and aged matched. Also the N for the natural infection is only 2 which is not really sufficient for making broad comparisons. It would have been much better to include a few additional samples. Is this possible? If not, the study is still informative but descriptive and comparisons should not be drawn requiring statistics as it is under-powered.

In terms of manuscript writing quality, the introduction has a bit too much results/discussion at the end of it and these points could be saved for the discussion section and streamlined a bit in the last paragraph of the intro. A little more information on the 45AZ5 challenge strain could be included in the introduction, not just based on the citation. It would be good to add the method of attenuation and the results of the Phase 1 study (still under consideration as a vaccine candidate?) if available.

For the “natural” infection group, were these not also examples of “controlled” infection? Presumably these patients survived/recovered with a normal disease course? There is not an indication in the data presented that they had higher viremia either.

Were the CD16hi monocytes the targets of infection? It is not clearly shown in the data. What percentage of the monocyte population were CD16hi monocytes? Is this the majority of the monocyte population or a subset of them?

This sentence from the discussion doesn’t make sense how the authors are drawing causation between these two things: “Our observation that monocytes appear to respond more dramatically to DENV-elicited systemic inflammation than B cells suggests that while monocytes are more permissive to infection in vitro, they become a much less attractive target of infection and replication in vivo.” Is this true that monocytes are less permissive in vivo and is it true that the responses relative to B cells give insight into this?

More information is needed on the statistical tests used in the “Statistical analysis” section of the manuscript.

Reviewer #3: 1. These experiments are expensive and difficult to do. However, the work is very descriptive and is limited in scope in terms of patient numbers with study groups of 3 and 2, making any comparisons very limited in scope.

2. Conclusions are made that natural primary DENV infection induces expression of IFNs and ISGs and that this is responsible for the suppression of mitochondrial function and cellular translation. However, from this study, this is impossible to say. There can be no causation implied from sequencing data. As the virus is known to manipulate mitochondrial function directly, it is possible that wild type DENV-1 is better able to do this resulting in the phenotypes.

3. Information on the attenuation of the experimental human DENV-1 construct is not provided in the text making it difficult to understand what the genetic background of this virus is.

4. No information is given on the infection status of individual cells. While the sequencing platform targets polyA, and flaviviruses are not thought to generate transcripts with polyA tails, studies by Mike Diamonds group were able to detect viral RNAs by scRNA-seq. Can the authors include or comment on this information.

**Part III – Minor Issues: Editorial and Data Presentation Modifications**

Reviewer #1: The manuscript reads as if different sections were written by different people. For example, the terms “DHIM”, “human challenge subset” “experimental DENV-1 infection” and “experimental DENV” are used interchangeably, even in figure panels. Similarly, wild-type infection is called “natural infection” and “primary infection” interchangeably. Please settle on one consistent term for each group.

Reviewer #2: The title should probably include the aspect of peripheral blood cells.

Reviewer #3: (No Response)

PLOS authors have the option to publish the peer review history of their article (what does this mean?). If published, this will include your full peer review and any attached files.

Reviewer #1: No

Reviewer #2: No

Reviewer #3: No
---

## [Editor Report · Decision Letter 1]

15 Dec 2020

Dear Dr Waickman,

We are pleased to inform you that your manuscript 'Temporally integrated single cell RNA sequencing analysis of PBMC from experimental and natural primary human DENV-1 infections' has been provisionally accepted for publication in PLOS Pathogens.

Best regards,

Daniela Weiskopf

Guest Editor

PLOS Pathogens

Ana Fernandez-Sesma

Section Editor

PLOS Pathogens

Kasturi Haldar

Editor-in-Chief

PLOS Pathogens

orcid.org/0000-0001-5065-158X

Michael Malim

Editor-in-Chief

PLOS Pathogens

orcid.org/0000-0002-7699-2064
---

## [Editor Report · Acceptance letter]

23 Jan 2021

Dear Dr Waickman,

We are delighted to inform you that your manuscript, "Temporally integrated single cell RNA sequencing analysis of PBMC from experimental and natural primary human DENV-1 infections," has been formally accepted for publication in PLOS Pathogens.

Best regards,

Kasturi Haldar

Editor-in-Chief

PLOS Pathogens

orcid.org/0000-0001-5065-158X

Michael Malim

Editor-in-Chief

PLOS Pathogens

orcid.org/0000-0002-7699-2064